# Hidden non-collinear spin-order induced topological surface states

Zengle Huang[1,4], Hemian Yi[2,4], Daniel Kaplan[1,3,4], Lujin Min[2], Hengxin Tan[3], Ying-Ting Chan[1], Zhiqiang Mao[2], Binghai Yan[3], Cui-Zu Chang[2] & Weida Wu[1] ✉

Rare-earth monopnictides are a family of materials simultaneously displaying complex magnetism, strong electronic correlation, and topological band structure. The recently discovered emergent arc-like surface states in these materials have been attributed to the multi-wave-vector antiferromagnetic order, yet the direct experimental evidence has been elusive. Here we report observation of non-collinear antiferromagnetic order with multiple modulations using spin-polarized scanning tunneling microscopy. Moreover, we discover a hidden spin-rotation transition of single-to-multiple modulations 2 K below the Néel temperature. The hidden transition coincides with the onset of the surface states splitting observed by our angle-resolved photoemission spectroscopy measurements. Single modulation gives rise to a band inversion with induced topological surface states in a local momentum region while the full Brillouin zone carries trivial topological indices, and multiple modulation further splits the surface bands via non-collinear spin tilting, as revealed by our calculations. The direct evidence of the non-collinear spin order in NdSb not only clarifies the mechanism of the emergent topological surface states, but also opens up a new paradigm of control and manipulation of band topology with magnetism.

Rare-earth monopnictides R(Bi,Sb) (R = Ce, Nd, Sm, etc.) with a rock salt structure have been extensively studied in the past 60 years for their intricate magnetic phase diagrams[1–4]. Except for the lanthanum, yttrium, and praseodymium compounds, most rare-earth monopnictides are antiferromagnets at low temperatures[1–4]. Upon cooling or applying a magnetic field, many of them undergo another or even multiple antiferromagnetic transitions[5–7], as a consequence of competing magnetic interactions. A notable example is the "Devil's staircase" in CeSb[8] and the magnetic transitions have a significant influence on the electronic band structures[9]. More recently, rare-earth monopnictides have been proposed to have topologically non-trivial band structures. A first-principle calculation predicted a systematic band inversion reduction from Ce to Yb compounds, and a topologically non-trivial to trivial transition occurs around SmSb and DyBi[10]. Later angle-resolved photoemission spectroscopy (ARPES) experiments confirmed the band inversion in rare-earth monobismuthides and identified linear-dispersed Dirac-like states[11–14]. Magnetotransport experiments revealed non-trivial Berry phase in SmBi[15] and SmSb[16], and possible Weyl or Dirac fermions in CeSb and NdSb when driving the samples to ferromagnetic phase with magnetic field or pressure[17–19]. Therefore, rare-earth monopnictides are an intriguing platform to explore the interplay between magnetism, strong electronic correlation, and topological band structure.

Remarkably, recent ARPES studies reveal that a number of rare-earth monopnictides, including NdBi, NdSb, and CeBi, host disconnected Fermi surface arcs emerging from an unconventional

[1]Department of Physics & Astronomy, Rutgers University, Piscataway, NJ 08854, USA. [2]Department of Physics, The Pennsylvania State University, University Park, PA 16802, USA. [3]Department of Condensed Matter Physics, Weizmann Institute of Science, Rehovot, Israel. [4]These authors contributed equally: Zengle Huang, Hemian Yi, and Daniel Kaplan. ✉e-mail: wdwu@physics.rutgers.edu

magnetic band splitting below Néel temperature ($T_N$)[20,21]. Initial density functional theory (DFT) calculations suggest that the emergent surface states cannot be explained by the ordinary A-type anti-ferromagnetism observed by neutron scattering experiment[3]. Later analysis suggests that multiple-wave-vector (multi-$q$) antiferromagnetic order[22] can reproduce the emergent surface states[23,24]. So far there is no direct experimental evidence of multi-q antiferromagnetic order, which is difficult to be differentiated from multi-domain state of simple $1q$ order in powder neutron diffraction[3]. Real-space magnetic imaging probes, such as spin-polarized scanning tunneling microscopy (SP-STM), are suitable for directly visualizing the non-collinear magnetic order[25,26].

In this work, we provide real space evidence of non-collinear antiferromagnetic order in NdSb using SP-STM. We observe a checkerboard magnetic contrast on the (001) surface of NdSb that is consistent with a non-collinear spin order, which is a superposition of two A-type magnetic orders with orthogonal in-plane modulations. Furthermore, we discover a hidden spin-rotation transition ($1q$-$2q$) 2 K below the Néel ordering temperature $T_N$ that has not been reported. More interestingly, the hidden transition coincides with the onset of band splitting observed by ARPES. DFT study reveals a band inversion driven by magnetic order-induced band folding, resulting in the unconventional magnetism-induced topological surface states in the family of rare-earth monopnictides.

## Results

Figure 1a shows an STM topographic image taken at 4.5 K with a nonmagnetic tip at sample bias $V_{bias} = +1.4$ V. The image reveals a square lattice of Nd atoms with Nd-Nd spacing ~4.3 Å calculated from the Bragg peaks $q_{lat} = (1, 0)$ and $(0, 1)$ in the Fourier transform map (Fig. 1c), in good agreement with prior studies[4]. The bias dependence of STM topography indicates the unoccupied states are dominated by Nd orbitals while the occupied states are predominantly of Sb orbitals, which is confirmed by our DFT calculations (Supplementary Fig. 2d). Interestingly, SP-STM topography in Fig. 1b reveals an additional checkerboard-like pattern at +1.4 V bias, resulting in two pairs of superlattice peaks of wave vectors $\mathbf{q}_{AFM} = (\pm\frac{1}{2}, \pm\frac{1}{2})$ as shown in Fig. 1d. The bias dependence of SP-STM topography shows that the checkerboard modulation is mainly visible in unoccupied states dominated by Nd orbitals, and gradually diminishes in occupied states with significant Sb character. The disappearance of checkerboard-like modulation is accompanied by a fractional $(\frac{1}{2}, \frac{1}{2})$ shift of the atomic lattice, presumably from Nd to Sb atoms (Supplementary Figs. 2–4). This further confirms that the observed checkerboard pattern comes from the spin-dependent tunneling between the magnetic tip and the local spin-polarized Nd orbitals.

Close inspection of the Fourier transform map (Fig. 1d) reveals that there is an asymmetry of the magnetic superlattice peak intensity along two orthogonal directions: the intensity of $\mathbf{q}_{AFM}^A = (\frac{1}{2}, \frac{1}{2})$ is stronger than that of $\mathbf{q}_{AFM}^B = (\frac{1}{2}, -\frac{1}{2})$. We also performed bias-dependent studies to confirm that asymmetry persists in the whole bias range when the magnetic modulation is visible (Supplementary Fig. 3). This magnetic asymmetry is uniform over hundreds of micrometers, indicating a global $C_4$ symmetry breaking. To confirm that this symmetry breaking is intrinsic, we examine the sample surface extensively over hundreds of micrometers and locate a magnetic domain boundary shown in Fig. 2c. The SP-STM image is taken at $V_{bias} = +0.3$ V to maximize the magnetic contrast. The magnetic

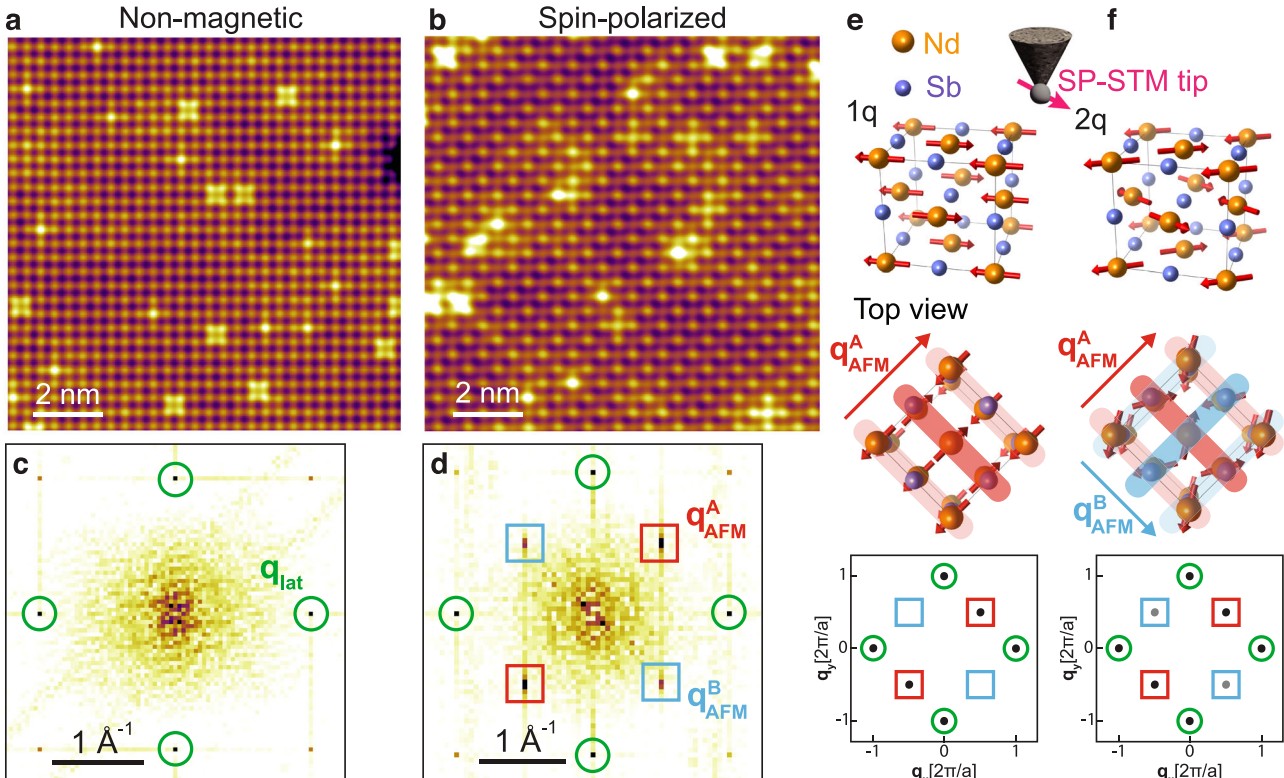

**Fig. 1 | STM and SP-STM topographic images of NdSb at 4.5 K. a, b** Non-spin-polarized and Spin-polarized topographic images of NdSb. Tunneling condition: $V_{bias} = +1.4$ V, $I = 1$ nA. **c, d** FFT of raw data of (**a**) and (**b**) respectively. The green circles highlight the lattice Bragg peak. The red squares highlight the magnetic Bragg peak. **e** top: The schematics of a spin-polarized STM tip scanning on the surface of NdSb with $1q$ spin structure; middle: the top view of the $1q$ spin structure. The red stripes illustrate the modulation along $\mathbf{q}_{AFM}^A$; bottom: the expected Fourier map of the $1q$ spin structure. **f** schematics of the $2q$ spin structure and expected Fourier map. The spins are not along the diagonal direction. It can be considered as the superposition of two unequal $1q$ modulations along $\mathbf{q}_{AFM}^A$ and $\mathbf{q}_{AFM}^B$. The resulting magnetic peaks are asymmetric.

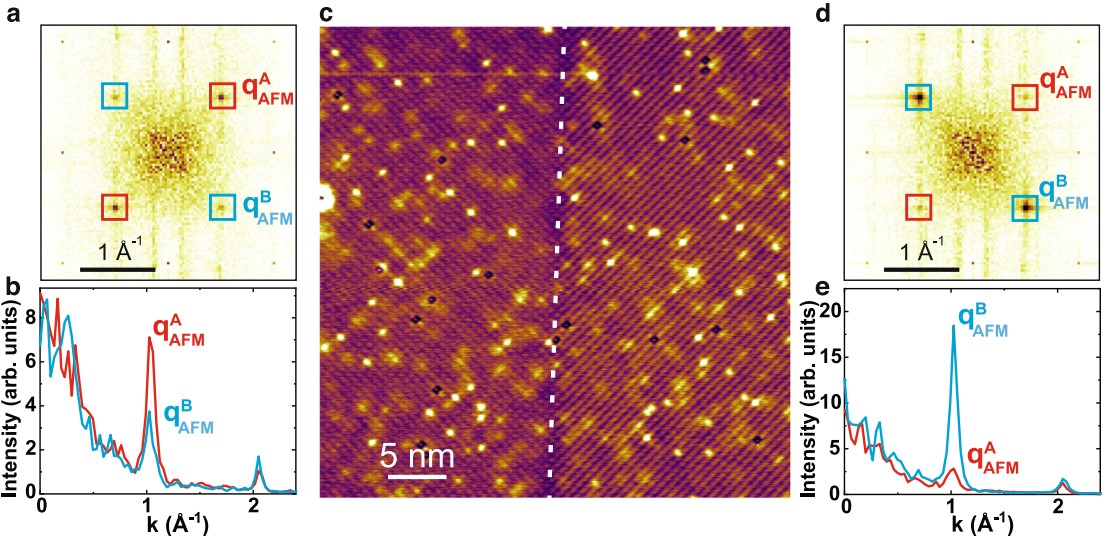

**Fig. 2 | Magnetic modulations across a domain boundary of NdSb. a** FFT obtained from the squared area on the left of the domain wall in **c**. $\mathbf{q}_{AFM}^{A}$ and $\mathbf{q}_{AFM}^{B}$ highlight the magnetic Bragg peaks along two orthogonal directions. **b** Line profiles along $\mathbf{q}_{AFM}^{A}$ and $\mathbf{q}_{AFM}^{B}$. **c** A spin-polarized topographic image with a magnetic domain wall highlighted by the white dashed line. Tunneling condition: $V_{bias} = 0.3$ V, $I = 0.3$ nA. **d**, **e** Similar to (**a**) and (**b**) but obtained from the squared area on the right of the domain wall in (**c**).

modulation asymmetry rotates 90° across the domain boundary, as clearly demonstrated in the Fourier transform maps of two domains and the corresponding line profiles along orthogonal directions (Fig. 2b, e).

This magnetic asymmetry suggests a magnetic structure that is different from that proposed in prior DFT analysis where the angle between Nd spin and cubic axes is either 0° ($1q$) or 45° (simple $2q$)[21,23,24]. The out-of-plane spin modulation in $3q$ cannot be detected by SP-STM, so we focus on the in-plane multi-wave-vector modulation and do not differentiate $2q$ and $3q$ in the following text. For the $1q$ magnetic structure, there is no magnetic modulation on the (001) surface, thus no magnetic superlattice peak. In contrast, there is a single modulation direction on the (110) [i.e., cubic (010)] surface, resulting in one pair of magnetic superlattice peaks (Fig. 1e). For the proposed simple $2q$ modulation[21,23], the Nd spin moments are 45° with respect to the cubic $a$, $b$-axes, so the 4-fold rotational symmetry (plus a fractional lattice translation) is restored. This magnetic modulation can be considered as a superposition of two orthogonal $1q$ modulations with equal spin components along the modulation directions, giving rise to two pairs of magnetic peaks with equal intensity on the (001) surface. Therefore, the two pairs of magnetic peaks with unequal intensity must stem from a magnetic structure with a spin configuration between $1q$ and the simple $2q$. In other words, the magnetic structure of NdSb is a $2q$ modulation with broken $C_4$ symmetry where the angle between Nd spin moments and the cubic $a$, $b$-axes is between 0° and 45°. In this scenario, the spin components along $\mathbf{q}_{AFM}^{A}$ and $\mathbf{q}_{AFM}^{B}$ are proportional to their modulation amplitudes, which are proportional to the magnetic peak amplitudes. Using the ratios of superlattice peak intensities, we estimate that the Nd moment rotates $\theta \approx 18°$ from $\mathbf{q}_{AFM}^{A}$. The angle is weakly temperature-dependent below 13 K (see Methods and Supplementary Fig. 10b).

To understand the origin of asymmetric magnetic modulation, we perform SP-STM measurements at various temperatures to examine the temperature dependence (Fig. 3b–d and Supplementary Information Figs. 5–7). At 11 K, the magnetic contrast is much weaker than that at 4.5 K, indicating a dramatic reduction of order moments. At around 14 K, the magnetic contrast is barely visible in SP-STM topography. From the Fourier transform the magnetic peaks with higher intensity ($\mathbf{q}_{AFM}^{A}$) remain while the ones with lower intensity ($\mathbf{q}_{AFM}^{B}$) disappear. Above $T_N (\approx 15$ K), all the magnetic superlattice peaks disappear,

indicating the recovery of the paramagnetic phase. Figure 3e shows the line profile of the Fourier transform intensity along two orthogonal directions crossing $\mathbf{q}_{AFM}^{A}$ and $\mathbf{q}_{AFM}^{B}$. Clearly, $\mathbf{q}_{AFM}^{B}$ vanishes above 13 K while $\mathbf{q}_{AFM}^{A}$ persists to $T_N$, indicating a $1q$ magnetic structure in this 2 K window. Therefore, there is a hidden $1q$-$2q$ transition at 13 K, which can be considered as anti-phase rotation of Nd spins in neighboring layers (Fig. 3a and Supplementary Fig. 11). So the $1q$-$2q$ transition is a spin-rotation transition. Just below $T_N$, the peak intensity ($\mathbf{q}_{AFM}^{A}$) is weak and develops slowly. However, it grows sharply below $T_R$ accompanied by the emergence of orthogonal modulation ($\mathbf{q}_{AFM}^{B}$), whose intensity also increases drastically. Therefore, our SP-STM results also suggest that the magnetic order parameter is greatly enhanced below spin rotation transition $T_R$.

The hidden spin-rotation transition provides new insight into the mechanism of the mysterious splitting of emergent surface states observed in NdBi and NdSb crystals[20,21]. In Fig. 4a, we plot the Fermi surface of NdSb measured by ARPES at 5.5 K. Consistent with previous results[21,24], the surface states are two-fold symmetric. In addition, the surface states also appear at the original $M$-point in the Brillouin zone of the paramagnetic phase. This observation agrees with prior ARPES results[14,24,27], confirming that the surface states are correlated with the band-folding induced by magnetic modulations[24,27]. Interestingly, two recent ARPES studies report an additional domain state with new 4-fold symmetric bulk Fermi surface in which the surface states are absent[27,28]. However, we didn't observe such a domain state in our NdSb crystal samples by either APRES or SP-STM measurements. Figure 4c shows the temperature evolution of band dispersion. Besides the two split surface bands (denoted as $\alpha$ and $\beta$) reported before[21], we observe a third surface band $\gamma$ in agreement with recent reports[14,24]. We extract the momentum distribution curves (MDCs) at the Fermi energy as shown in Fig. 4e. Above $T_N$, only a broad shoulder-like feature due to the bulk bands can be observed. Below $T_N$, a peak appears as a signature of the emergent surface states. The intensity of the peak increases gradually upon cooling but it does not split into $\alpha$ and $\beta$ bands until below 13 K, in good agreement with a recent APRES study[27]. Interestingly, the onset temperature of surface band splitting coincides with $T_R$, further substantiating that the spin-rotation transition is an intrinsic phenomenon of NdSb. Note that the same behavior of

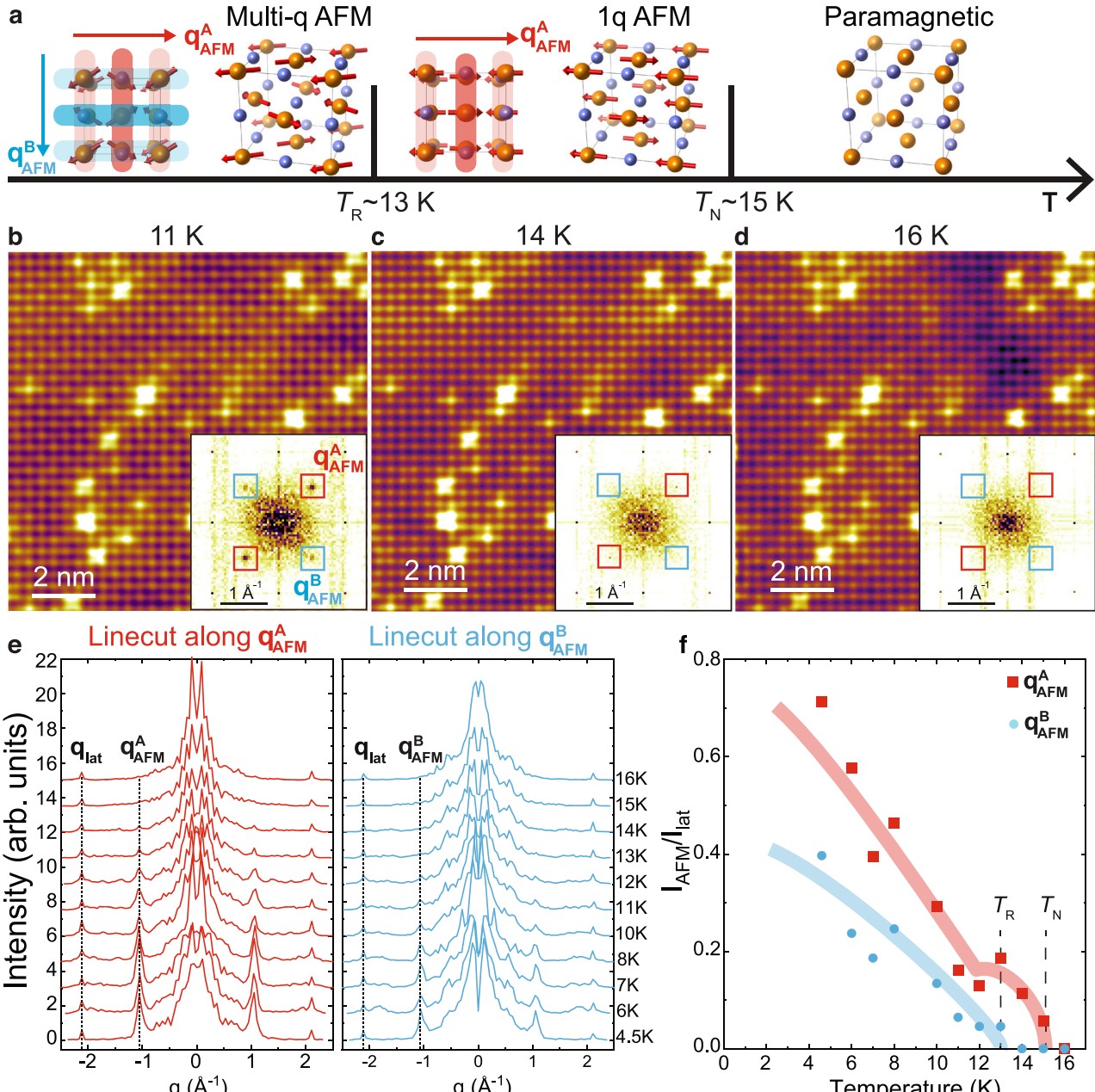

**Fig. 3 | Temperature-dependence of magnetic modulations in NdSb. a** The magnetic phase diagram in NdSb. **b–d** Spin-polarized topographic images taken at the same area at 11 K (**b**), 14 K (**c**) and 16 K (**d**). The insets are the corresponding FFT and the positions of the magnetic Bragg peaks are highlighted by the red ($\mathbf{q}_{AFM}^{A}$) and blue ($\mathbf{q}_{AFM}^{B}$) squares. **e** The temperature-dependent line profiles along $\mathbf{q}_{AFM}^{A}$ (red) and $\mathbf{q}_{AFM}^{B}$ (blue) magnetic Bragg peaks. **f** The normalized magnetic Bragg peak intensity of $\mathbf{q}_{AFM}^{A}$ and $\mathbf{q}_{AFM}^{B}$ as a function of temperature. The blue and red lines are guides to the eyes.

surface band splitting below $T_N$ ($\approx 24$ K) has been observed in the sister compound NdBi, where the emerging surface states split below 20 K[20]. This indicates a similar hidden spin-rotation at in NdBi, which deserves a future exploration. Our results suggest that the hidden transition is likely a universal phenomenon for the Nd monopnictides.

To understand the fundamental mechanism of emergent surface states and their splitting, we perform DFT calculations to understand the bulk and surface states evolution from PM to $1q$ and from $1q$ to $2q$ transitions. The PM phase of NdSb is topologically trivial because of the lack of direct band inversion between the Nd 5 $d$ and Sb 3 $p$ orbitals (Supplementary Figs. 13a, b and 14a). In contrast, the $1q$ or $2q$ phase leads to a band folding along $\Gamma - X$ in the bulk, resulting in the band inversion between the Nd 5 $d$ and Sb 3 $p$ orbitals (Supplementary

Fig. 14b). Here, spins align along the $x$-direction in the $1q$ phase. The reduced symmetry also allows the opening of a topological band anti-crossing gap near the Fermi energy ($E_F$). Consequently, two surface bands emerge from the upper and lower bulk band edges involved in the anti-crossing gap and disperse in the direction of $\Gamma - X$ (Fig. 4d). The surface band splitting is small in $1q$ phase because of the small anti-crossing gap.

Because of multiple band crossings at different energies and momenta in the full Brillouin zone, the global topological invariant of NdSb is trivial even in the antiferromagnetic phases. However, this does not necessarily negate the existence of topological surface states near $E_F$ in the local momentum region around $\Gamma$, which is distinct from the case of a trivial insulator as shown in Supplementary Fig. 13a, b.

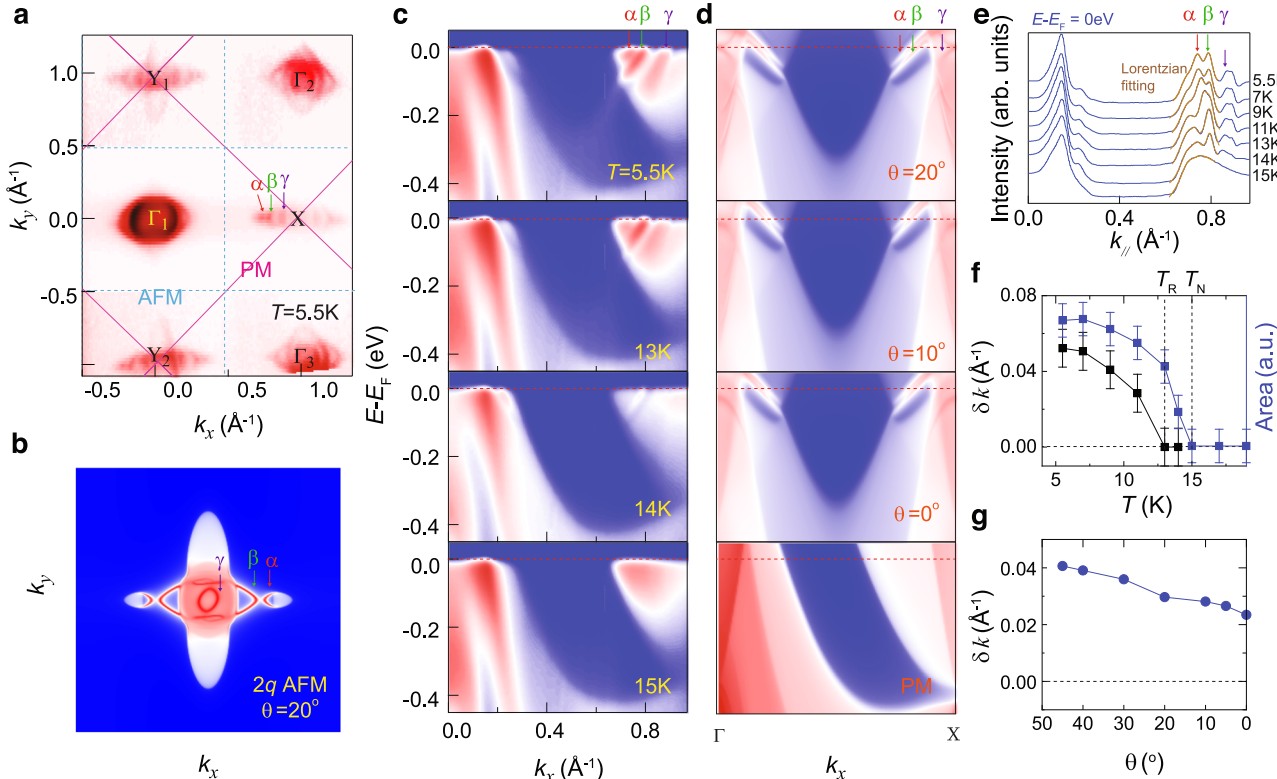

**Fig. 4 | ARPES spectra at different temperatures and DFT calculated band structure of NdSb (001) surface. a** Fermi surface map of NdSb (001) measured at $T = 5.5$ K. The photon energy of ~53 eV is used. The purple solid lines outline the Brillouin zone for the paramagnetic phase of NdSb. The blue dashed lines outline the folded Brillouin zone for the multi-$q$ antiferromagnetic phase of NdSb. $\alpha$, $\beta$ and $\gamma$ denote three emergent surface states. **b** DFT calculated Fermi surface of $2q$ antiferromagnetic phase projected on NdSb (001) surface. **c** Temperature evolution of ARPES spectra of NdSb along the $\Gamma X$ direction. **d** DFT calculated $\Gamma X$ band

structure of NdSb at varying angles of Nd spin moments. **e** Temperature-dependence of the MDCs at the Fermi level, as marked by a red arrow in (**c**). The peaks of $\alpha$ and $\beta$ surface states are fitted by the Lorentzian function. **f** Temperature-dependence of the MDC peak splitting (black) and the sum of peaks area for the $\alpha$ and $\beta$ surface states (blue). The error bars are obtained from the Lorentzian fittings in (**e**). **g** The calculated band splitting of $\alpha$ and $\beta$ surface state as a function of the angle between Nd spin moments and $a$, $b$-axes.

Below the $1q$-$2q$ transition, spins rotate in the $xy$ plane to form a non-collinear order. Such spin rotation increases the anti-crossing gap and thus splits the surface bands further (Fig. 4d, g, and Supplementary Fig. 12) according to our DFT calculations, which correspond to the zero temperature limit. In finite temperatures, the split is expected to be reduced because thermal fluctuations suppress the magnitude of the order moment. Consistently, the observed temperature dependence of surface band splitting from ARPES results (Fig. 4f) follows that of spin-modulation amplitude in Fig. 3f, which is a measure of the order moment size. Near and above the $1q$-$2q$ transition, the spin-modulation amplitude is so weak that the surface band splitting is smaller than peak width, resulting in a single surface band observed in ARPES measurements[20,27]. In other words, the surface band splitting below the $1q$-$2q$ transition is due to the enhancement of the anti-crossing gap in the non-collinear antiferromagnetic state ($2q$) because of an enhancement of order moment. Besides understanding the origin of surface band splitting, our DFT calculations also confirm that the Fermi surface projected on the (001) surface of $2q$ phase with rotation angle ($\theta = 20°$) similar to the experimental value is two-fold symmetric (Fig. 4b), in good agreement with the ARPES results shown in Fig. 4a. Therefore, the ground state of NdSb is a $2q$ antiferromagnetic structure with broken $C_4$ symmetry.

The combined SP-STM, ARPES, and DFT studies of NdSb establish that the band folding due to antiferromagnetic ordering induces topological band inversion at Brillouin zone boundaries, resulting in emergent topological surface states. Furthermore, our studies unveil a hidden spin-rotation transition from $1q$ to $2q$ spin orders 2 K below $T_N$,

which is responsible for the enhanced order moment and the resultant unconventional magnetic band splitting. The non-collinear spin order and hidden spin-rotation transition might also exist in other rare-earth monopnictides. Besides clarifying the mechanism of topological surface states, our work provides an example of magnetism-induced topological phase transition, thus enabling an exciting possibility of tuning band topology with magnetism.

## Methods

### STM and SP-STM measurements

The STM measurements were carried out in an Omicron LT-STM with base pressure better than $1 \times 10^{-11}$ mbar. For non-spin-polarized STM, electrochemically etched tungsten tips were conditioned on a clean Au (111) surface before experiments. For SP-STM experiments, the spin-polarized tip is functionalized by repeatedly scanning the nonmagnetic tungsten tip on cleaved surfaces of FeTe single crystals[29,30]. The spin-polarized tip is confirmed by visualizing the in-plane collinear antiferromagnetic order on FeTe (Supplementary Fig. 1). NdSb single crystals were cleaved in-situ around 5 K and then immediately inserted into the cold STM head. All STM images in the paper are drift-corrected using the Lawler-Fujita algorithm[31]. High-frequency noise was removed by the low-passed filter in the Fourier maps (Supplementary Fig. 5).

### ARPES measurements

The ARPES measurements were carried out at Beamline 10.0.1, Advanced Light Source, Lawrence Berkeley National Laboratory. A fresh NdSb (001) surface was achieved by cleaving the NdSb crystal at

$T = 20$ K. One surface termination was exposed for the cubic structure of NdSb, similar to other Rare-earth monopnictides. The base vacuum of the ARPES chamber is better than ~$5 \times 10^{-11}$ mbar. A hemispherical Scienta R4000 analyzer was used in our ARPES measurements. The energy and angle resolution was set to ~15 meV and ~0.1°, respectively. The spot size of the photon beam is ~$100 \times 100\ \mu m^2$.

## DFT calculations

We carried out DFT calculations using the Vienna ab-initio software package (VASP)[32,33]. The lattice constant adopted for NdSb is $a = 6.70$ Å, in order to match the ARPES finding that the paramagnetic phase is trivial. The properties of Nd $f$ electrons were corrected with a Hubbard $U = 6.3$ eV and $J = 0.7$ eV using Dudarev's method[34]. The plane wave states were then projected onto a tight-binding basis of Nd $d, f$ and Sb $p$ orbitals using Wannier90[35]. Surface states were calculated using efficient, iterative Green's function methods as implemented in WannierTools[36].

## NdSb single crystal growth

NdSb single crystals were synthesized by a Sn flux method[37]. High-quality powders of Nd, Sb, and Sn in a ratio of 1:1:20 were put into an alumina crucible which was subsequently sealed within a vacuumed quartz tube. The prepared ampoule was heated up to 1150°C and then slowly cooled down to 750°C at a rate of 3°C/h, followed by decanting of the excess liquid using a centrifuge. Shining cubic crystals were obtained after breaking the crucible.

## Density of states for Nd and Sb

In Supplementary Fig. 2b, c, we show the non-spin-polarized STM image taken at the tunneling bias of $+1.4$ V and $-1.3$ V. It is evident that at $+1.4$ V the highlighted defects are centered on the protrusions whereas at $-1.3$ V the depressions, indicating at $+1.4$V and $-1.3$ V different atoms (Nd or Sb) are imaged. In Supplementary Fig. 2d We plot the ab-initio calculated partial density of states, projected on the atomic sites in the $2q$ configuration of NdSb. We focus on the energy window probed by STM with the Fermi level set to zero. For negative voltages, the density of states is dominated by Sb $p$ orbitals, while for positive voltages the density of states is composed primarily of Nd $d, f$ orbitals. This suggests that at positive tunneling biases, the bright atom-like features are Nd atoms because of their large DOS above Fermi energy, while Sb atoms are imaged at negative bias.

## Estimation of Nd spin moment angle from spin-polarized STM images

In Fig. 2 and Supplementary Fig. 8, we observe a magnetic domain wall separating two magnetic domains of different magnetic contrasts. The amplitude asymmetry of the magnetic peaks $\mathbf{q_A}$ and $\mathbf{q_B}$ in the Fourier transform switches sign across the domain wall. Therefore, we can estimate the angles between Nd spin moments and the magnetic modulation using the magnetic peak intensities of these two domains.

Supplementary Fig. 10a shows the schematics of AFM modulations ($\mathbf{q_A}$ and $\mathbf{q_B}$), tip moments (green arrows) and Nd moments (orange arrows) on the Nd lattice (black dots). On the left domain, we denote the angle between Nd moment and $\mathbf{q_A}$ as $\theta$, and the angle between the tip moment and $\mathbf{q_A}$ as $\alpha$. As explained in the main text, the $2q$ modulation can be decomposed into two unequal $1q$ modulations along $\mathbf{q_A}$ and $\mathbf{q_B}$, whose spin moments are along the modulation wave vectors. The sum of the spin moments of these two $1q$ modulations gives rise to the spin moments in $2q$. Without losing generality (e.g. in Supplementary Fig. 8), we assume modulation amplitudes $S_A^L = S\cos\theta$ are stronger than $S_B^L = S\sin\theta$ on the left domain, i.e., assuming $S_A^L > S_B^L$. Thus, $\theta < 45°$ by definition. The situation is the opposite on the right domain, i.e., $S_A^R = S\sin\theta$ and $S_B^R = S\cos\theta$. Therefore, the Nd moments on the left domain are more inclined to $\mathbf{q_A}$, and vice versa for the right domain. Therefore, the Nd moment angle $\theta$ is related to the ratio of the modulation amplitudes $S_A$ and $S_B$ along $\mathbf{q_A}$ and $\mathbf{q_B}$ through the following equation:

$$\frac{S_A}{S_B} = \frac{S\cos\theta}{S\sin\theta} = \frac{1}{\tan\theta} \tag{1}$$

In SP-STM, the spin-dependent component of tunneling current is proportional to the projection of local spin orientation onto the tip moment[38]. Therefore, the measured magnetic peak amplitudes with SP-STM are determined by projections of the two $1q$ modulations onto the tip moment.

For the left domain:

$$\frac{I_A^L}{I_B^L} = \frac{S_A^L\, m_{tip}\, \cos\alpha}{S_B^L\, m_{tip}\, \sin\alpha} = \frac{S\cos\theta\cos\alpha}{S\sin\theta\sin\alpha} = \frac{1}{\tan\theta \cdot \tan\alpha} \tag{2}$$

For the right domain:

$$\frac{I_A^R}{I_B^R} = \frac{S_A^R\, m_{tip}\, \cos\alpha}{S_B^R\, m_{tip}\, \sin\alpha} = \frac{S\sin\theta\cos\alpha}{S\cos\theta\sin\alpha} = \frac{\tan\theta}{\tan\alpha} \tag{3}$$

where $I_A^L, I_B^L, I_A^R, I_B^R$ are the magnetic peak amplitudes along $\mathbf{q_A}$ and $\mathbf{q_B}$ directions obtained from the Fourier transform of the left and right domains in SP-STM images, respectively. We use the following two relations to compute $\alpha$ and $\theta$:

$$\begin{cases} \theta = \arctan\left(\sqrt{\dfrac{I_A^R}{I_B^R} \cdot \dfrac{I_B^L}{I_A^L}}\right) \\[4mm] \alpha = \arctan\left(\sqrt{\dfrac{I_B^R}{I_A^R} \cdot \dfrac{I_B^L}{I_A^L}}\right) \end{cases} \tag{4}$$

The temperature-dependence of the angle $\theta$ between the Nd spin moments and the in-plane $a, b$-axes is plotted as orange squares in Supplementary Fig. 10b. The angle is around 18° from 4.5 K up to 12 K ($\lesssim T_R$).

## Topological features of the 1q to 2q transition

To verify the topological nature of the surface states, we connect the theoretically calculated and experimentally observed states with bulk band crossings. In our calculations of the non-magnetic phase of NdSb, band structure and surface spectrum shown in Supplementary Fig. 13a, b, is topologically trivial with no surface states, consistent with Fig. 4c for $T \geq T_N$. Upon the onset of magnetic order, the $1q$ phase assumes the magnetic space group P4/mm'm'. For this configuration, folding due to magnetism occurs only along the axis *parallel* to the magnetic moment direction, while there is no folding along the directions perpendicular to the magnetic moment axis. Therefore, we find surface states along $\bar{\Gamma} - \bar{X}$, but not along $\bar{Y} - \bar{\Gamma}$. As shown in Supplementary Fig. 12a–g, the appearance of surface states is matched by the opening of a gap in the bulk near $E_F$ (dashed line). This indicates the topological origin of the states which result from the magnetic order-induced folding. When such folding is absent (or trivial), as along $\bar{Y} - \bar{\Gamma}$, surface states do not appear. When the spins begin to rotate towards the $2q$ configuration, the symmetry is reduced further, and the magnetic space group is given by Pc'c'm' for all $0 < \theta < 45°$. This induces folding along all axes. The emergence of these surface states is always accompanied by a bulk band inversion and gap opening, which manifests in the appearance of surfaces states which merge with the bulk at finite $k$. It should be noted that these states are only present as a result of the folding induced by the magnetic ordering. The splitting of the surface states along $\bar{\Gamma} - \bar{X}$ is also strongly tied to the magnetic order. We take a line cut along $\bar{\Gamma} - \bar{X}$ for $E = 0.07$ eV and plot the momentum splitting of the two surface states corresponding to the $\alpha, \beta$ bands in Fig. 4g. As the spins are rotated away from the $1q$ configuration, the splitting increases, as the symmetry is reduced further. Direct evidence for the band inversion can be inferred from the band characters near $E_F$. We calculate the

irreducible representations of $C_{4x}$ ($x$ being the spin direction) along $\Gamma - X$. We find a band inversion (band character inversion) precisely at the gap opening in the bulk band structure. This is illustrated in Supplementary Fig. 13c. The total topological indices of the system (calculated at the TI-invariant momenta) remain zero, as in the paramagnetic state. For simplicity and continuity of illustration, all band dispersions are plotted in terms of the folded unit cell.

## Data availability

The spin-polarized STM data presented in the main text and supplementary information have been deposited in the Dryad database https://doi.org/10.5061/dryad.280gb5mv3[39]. All ARPES data and DFT calculations needed to evaluate the conclusions in the paper are present in the paper and/or the Supplementary Information. Additional data related to this paper are available from the authors upon request.

## Code availability

The code for STM data analysis has been deposited in the Dryad database https://doi.org/10.5061/dryad.280gb5mv3[39]. The codes for ARPES analysis and the DFT calculations are available from the authors upon request.

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

## Acknowledgements

We thank Kristjan Haule for the helpful discussions. We thank Jiaqiang Yan for providing FeTe single crystals. The SP-STM measurements (Z.H. and W.W.) at Rutgers were supported by the Office of Basic Energy Sciences, Division of Materials Sciences and Engineering, U.S. Department of Energy under Award No. DE-SC0018153. W.W. also acknowledges the support from the Army Research Office grant (No. W911NF-20-1-0108). The ARPES measurements (H.Y. and C.-Z.C) and the crystal growth (L.J.M. and Z.Q.M.) were supported by the Penn State MRSEC for Nanoscale Science (DMR-2011839). The DFT calculations (D.K., H.T., and B.Y.) were supported by the European Research Council (ERC Consolidator Grant "NonlinearTopo", No. 815869). C.-Z.C also acknowledges the support from the NSF grant (DMR-2241327) and the Gordon and Betty Moore Foundation's EPiQS Initiative (Grant No. GBMF9063 to C.- Z.C.). Z.Q.M. also acknowledges the support from NSF under Grants No. DMR 2211327.

## Author contributions

H.Y., C.-Z.C. and W.W. initiated the project. Y.-T.C., Z.H. and W.W. conceived and designed the SP-STM experiments. Z.H. performed the STM and SP-STM measurements and analyzed the STM data. H.Y. and C.-Z.C. performed ARPES measurements and data analysis. D.K., H.T. and B.Y. performed the DFT calculations and theoretical analysis. L.J.M. and Z.Q.M. grew NdSb single crystals. The manuscript was written by Z.H., H.Y., D.K., B.Y., C.-Z.C. and W.W. All authors discussed and commented on the manuscript.

## Competing interests

The authors declare no competing interests.
