## [Peer Review File · Nature Communications]

Reviewers' Comments:

Reviewer #1:

Remarks to the Author:

The work by Z. Huang and coauthors studied the spin order in NdSb. By applying the spin polarized STM, the authors provide the evidence that the NdSb possesses the non-collinear spin order. As summarized in Fig. 1 and Fig. 2, the 2q anti-ferromagnetic order in NdSb is clearly shown in the FFT image. The authors further measured the temperature dependent evolution of the spin order, and clearly showed the phase transition in Fig. 3: the transition from the paramagnetic state to the 1q antiferromagnetic state occurs at about 15K, and the transition from the 1q antiferromagnetic state to the 2q antiferromagnetic state occurs at about 13K. With the 2q anti-ferromagnetic order observed, the authors performed the DFT calculations to show how the 1q-2q transition affect the band structure of NdSb.

As the authors stated in the introduction, there is no direct experimental evidence of the 2q anti-ferromagnetic order in the rare-earth monpnictides. The authors' spin resolved tomographic image clearly showed the 2q anti-ferromagnetic order in NdSb and also demonstrated how the spin order evolves as the temperature goes down, which makes their work stand out. Based on the reasons above, I would like to recommend their work to get published in Nat. Commun. However, before the publication of their work, I hope the authors to address a few issues raised in the below:

1) the q^A_{AFM} order and q^B_{AFM} order seems to be quite symmetric, and they can appear individually in each side of a domain wall, as is indicated in Fig. 2. Is it true that the q^A_{AFM} order and the q^B_{AFM} order is connected by some crystalline symmetry, like the mirror or C_4 ? If they are crystalline symmetry connected, they should occur in the same temperature. What is the reason that an order with q^A_{AFM} occurs first and then comes the q^B_{AFM} order? Is it due to some slight symmetry breaking of the sample induced by strain (or other factors) that lifts the energy degeneracy of the q^A and q^B spin order?

2) In the work the authors mentioned that the magnetic order induced band folding can give rise to a band inversion and further induce some topological surface states. Usually we suppose that a band inversion is induced by a spin orbit coupling. What is the reason that the spin order observed here can induce a band inversion?

Reviewer #2:

Remarks to the Author:

See attached.

Reviewer #3:

Remarks to the Author:

Huang et al. have conducted interesting experiments on a rare-earth monpnictide NdSb which is believed to exhibit complex magnetism interplaying with a topological band structure. The authors observed non-collinear antiferromagnetic order with multiple modulations using SPSTM. They also discovered a hidden spin-rotation transition which seems to be related to the splitting of surface states measured in ARPES. Using DFT calculations, the authors showed the magnetic orderings induce band folding in Brillouin zone, leading to the band gap opening in the inverted bulk band which provides room for the existence of the topological surface states. This manuscript establishes an interesting example of magnetism interplaying with topology in real materials, so I believe the manuscript has the potential for publication in Nature Communications. However, I have several questions that are not entirely clear to me within the present manuscript.

(1) How can the 1q-spin structure (Fig. 1e) result in the strip pattern in the SPSTM measurement? If I consider the spins of the top Nd atoms, I am not sure why q_b_{AFM} peaks should be absent. Should I take into account the underlying Nd atoms or the top Sb atoms to understand the missing q_b_{AFM} peaks? If so, I don't agree with the authors regarding STM's ability to detect the spins of

buried Nd atoms.

(2) What causes the anti-crossing in the folded bulk bands? Is it spin-orbit coupling? How the spin-rotation increases the anti-crossing gap? How does the increased anti-crossing gap split the surface band further?

(3) What drives the spin-rotation transition around 13 K in terms of energy?

(4) In Fig. 4a, it's unclear why the ARPES intensities are not repeated within the AFM Brillouin zone (indicated by blue dashed lines) if the Brillouin zone is folded in the AFM phase of NdSb. Can the authors provide a demonstration of how the bulk bands in the paramagnetic phase fold in the AFM phase? This clarification could be beneficial for understanding the mechanisms of band inversions as a result of Brillouin zone folding.

(5) In Fig. 4, NbSb should be NdSb in several places.

Point-by-point Responses

The reviewers' comments are in black font.

Our replies are in blue font.

Reviewer Comments

Reviewer #1 (Remarks to the Author):

The work by Z. Huang and coauthors studied the spin order in NdSb. By applying the spin polarized STM, the authors provide the evidence that the NdSb possesses the non-collinear spin order. As summarized in Fig. 1 and Fig. 2, the 2q anti-ferromagnetic order in NdSb is clearly shown in the FFT image. The authors further measured the temperature dependent evolution of the spin order, and clearly showed the phase transition in Fig. 3: the transition from the paramagnetic state to the 1q antiferromagnetic state occurs at about 15K, and the transition from the 1q antiferromagnetic state to the 2q antiferromagnetic state occurs at about 13K. With the 2q anti-ferromagnetic order observed, the authors performed the DFT calculations to show how the 1q-2q transition affect the band structure of NdSb.

As the authors stated in the introduction, there is no direct experimental evidence of the 2q anti-ferromagnetic order in the rare-earth monpnictides. The authors' spin resolved tomographic image clearly showed the 2q anti-ferromagnetic order in NdSb and also demonstrated how the spin order evolves as the temperature goes down, which makes their work stand out. Based on the reasons above, I would like to recommend their work to get published in Nat. Commun. However, before the publication of their work, I hope the authors to address a few issues raised in the below:

Reply: We thank the reviewer for a nice summary and support for publication of our work. Below we provide point-to-point replies to his/her comments/questions.

1) the q^A_{AFM} order and q^B_{AFM} order seems to be quite symmetric, and they can appear individually in each side of a domain wall, as is indicated in Fig. 2. Is it true that the q^A_{AFM} order and the q^B_{AFM} order is connected by some crystalline symmetry, like the mirror or C_4 ? If they are crystalline symmetry connected, they should occur in the same temperature. What is the reason that an order with q^A_{AFM} occurs first and then comes the q^B_{AFM} order? Is it due to some slight symmetry breaking of the sample induced by strain (or other factors) that lifts the energy degeneracy of the q^A and q^B spin order?

Reply: The reviewer is correct that q^A and q^B are related by C_4 . The C_4 symmetry is broken below T_N due to a concomitant structural distortion that is coupled to the 1q AFM state (Please see Phys. Kondens. Mater. **10**, 85 (1969) and Ref. [4] (J. Phys. C: Solid State Phys. 6, 725-737 (1973)),

as the reviewer correctly guesses. The broken C_4 symmetry makes q^A and q^B inequivalent so that only one of them appears below T_N .

2) In the work the authors mentioned that the magnetic order induced band folding can give rise to a band inversion and further induce some topological surface states. Usually we suppose that a band inversion is induced by a spin orbit coupling. What is the reason that the spin order observed here can induce a band inversion?

Reply: The reviewer is correct that topological band inversion is usually induced by the spin orbit coupling and it is indeed the case here. Please see Figure R1 for the demonstration of the band structure and band folding from our DFT calculations. In the paramagnetic phase, a Nd-5d band minimum at the X point is lower than a Sb-3p band maximum at the Γ point, because of the spin-orbit coupling. Below Neel temperature, the magnetic order folds the Nd-5d band to the Γ point, making it cross with the Sb-p band, resulting in an inversion of effective direct band gap. Thus, the spin order and the resulting symmetry reduction lead to the anti-crossings and open a topological hybridization gap at the crossing points.

We added Figure R1 to as Supplementary Fig.14 in the Supplementary Information and added corresponding discussion in the revised manuscript to clarify the band folding and the topological hybridization gap opening.

Figure R1. Schematics of the band structure in the PM/Folded PM/AFM phase of NdSb. (a) Bulk band structure in the PM phase of NdSb, without folding. (b) Bulk band structure of the PM phase of NdSb, when trivially folded. The dashed red lines indicate the folded bands, when the high symmetry X point is folded onto the Gamma point, $X \rightarrow \Gamma$. Note that crossings are now enabled, between Nd-d and Sb-p orbitals. The lack of magnetic order prevents a gap opening (due to the higher cubic symmetry present), as the folding is due to the choice of a different unit cell. (c) When magnetic order sets on and the symmetry is reduced, anti-crossings are allowed. The topological nature of the gap is confirmed by the change in the band character under the representation of rotational symmetry. This process is also illustrated in Supplementary Fig. 13. (d) A simplified cartoon showing the band folding between the bands of Nd-d and Sb-p characters and the magnetic-order-induced topological gap opening.

Reviewer #2 (Remarks to the Author):

In this manuscript, the authors have performed spin-polarized scanning tunneling microscopy (SP-STM), DFT, and ARPES studies on a rare-earth monpnictide material NdSb which has gained lot of attention recently due to the observation of arc-like states in its electronic band structure. They have provided evidence of non-collinear antiferromagnetic (AFM) order using SP-STM. In addition to this, they have also discovered a spin-rotation transition around 2K below the Neel temperature. They have observed that this spin-rotation transition temperature coincides with the onset of band splitting observed in ARPES. The manuscript is well-written, and their findings from SP-STM demands publication, however, I feel that this manuscript is not at the level of Nature Communication journal as the work lacks novelty. This manuscript fits better for a more specialized journal. I will provide my comments below.

Reply: We thank the Reviewer for concise summarizing our manuscripts and highlighting our manuscript well-written. However, we disagree respectfully with the reviewer’s comment that our work “lacks novelty”. We would like to point out that the novelty of our manuscript has been recognized by Reviewer #1 and Reviewer #3. Reviewer #1 states that “*The authors’ spin resolved tomographic image clearly showed the 2q anti-ferromagnetic order in NdSb and also demonstrated how the spin order evolves as the temperature goes down, which makes their work stand out.*” Reviewer #3 states: “*This manuscript establishes an interesting example of magnetism interplaying with topology in real materials, so I believe the manuscript has the potential for publication in Nature Communications.*” Thus, we believe our work is commensurate with the high standards of Nature Communications. Below we provide point-to-point replies to his/her comments/questions.

1. In page 6, the authors state in the second paragraph, “In addition, our ARPES unveils a larger momentum space and the surface states also emerge at the original M-point in the Brillouin zone of the paramagnetic phase”. Further they state that they have observed a third surface band □ which is not reported in prior studies and cite the following paper [Nature 603, 610-615 (2022),

Phys. Rev. B 106, 115112 (2022)]. The ARPES data presented by the authors in Fig. 4 is not novel. Similar study has already been reported in the paper [Phys. Rev. B 106, 235119 (2022)]. The authors should discuss how the present ARPES data adds to the already published paper in [Phys. Rev. B 106, 235119 (2022)]. These arc-like states have been further discussed in details in another paper published in *npj Quantum Mater.* 8, 22 (2023). The authors have presented Fig. 4a,c as a new discovery in the present manuscript where they try to highlight that their present study extends the previous study [Nature 603, 610-615 (2022), Phys. Rev. B 106, 115112 (2022)] and cover a larger part of the Brillouin Zone. This is not true and the authors should cite the papers [Phys. Rev. B 106, 235119 (2022), *npj Quantum Mater.* 8, 22 (2023)] when they discuss ARPES data in Figure 4. They should further discuss what novel features they observe in ARPES apart from what has already been reported by these papers.

Reply: We thank the Reviewer for highlighting the reference papers [that is Schruck et. al, *Nature* 603, 610-615 (2022), Kushnirenko et. al, *Phys. Rev. B* 106, 115112 (2022), and Li et. al, *npj Quantum Mater.* 8, 22 (2023)] and bringing the paper [Sakhya et. al, *Phys. Rev. B* 106, 235119 (2022)] to our attention. As recognized by other reviewers, it is the discovery or hidden spin-rotation and its connection to the topological nature of surface states that make our work stands out. Our ARPES results are one of important components of our work.

We cited [Schrunk et. al, *Nature* 603, 610-615 (2022), Kushnirenko et.al, *Phys. Rev. B* 106, 115112 (2022), and *npj Quantum Mater.* 8, 22 (2023)] in our original manuscript [Ref. 20, 21, 24]. We agree with reviewer #2 that “*Similar study has already been reported in the paper [Phys. Rev. B 106, 235119 (2022)].*” Sakhya et. al reported three extra features of ARPES spectra at X point in the antiferromagnetic phase of NdSb. These extra electronic features are the same as the surface states observed in our ARPES spectra in original M-point (denoted X point in Sakhya’ paper) in the Brillouin zone of the paramagnetic phase. Moreover, we also observed surface states near the G point in the first BZ. With the first principles calculations, we found that the surface states near the M point are the band folding of surface states near G due to the formation of antiferromagnetic order. Our observation of the formation of surface states and their folding effect under the antiferromagnetic phase is also seen in two very recent papers [Fig. 2d in Li et. al, *npj Quantum Mater.* 8, 22 (2023) and Fig. 3c in Honma et. al, *Phys. Rev. B* 108, 115118 (2023)]. We note that Honma’s ARPES work was recently published during the evaluation and review process for our manuscript. In summary, our APRES results agree well with existing literatures.

Besides the excellent agreement with prior works, our ARPES results reveal the temperature dependence of the surface band splitting at near M point due to the antiferromagnetic phase transition. The onset of the surface state splitting occurs 2 K below the Néel temperature (Fig. 4e, f in main text). These fine electronic structure features further support the spin-rotation transition observed by our spin-polarized STM. It is worth noting that a similar temperature evolution of surface band splitting was observed near the G point in a very recent paper [Fig. 3b in Honma et.al, *Phys. Rev. B* 108, 115118 (2023)]. This is further evidence for the folding effect of surface states and spin-rotation transition in the antiferromagnetic phase of NdSb. In our revised manuscript, we cited the aforementioned two papers [Sakhya et. al, *Phys. Rev. B* 106, 235119 (2022), Honma et. al, *Phys. Rev. B* 108, 115118 (2023)] and add the related discussion on the ARPES results.

2. The authors claim that the band folding due to 2q antiferromagnetic modulation induces topological band inversion at BZ boundaries, resulting in emergent topological surface states. How do the authors confirm these are topological surface states from their experiments?

Reply: We thank the Reviewer for bringing up this question. As we discussed in our manuscript, our first-principles calculations reveal the origin of the topological surface states and the temperature dependence of the surface band splitting due to the formation of antiferromagnetic phase transition. By performing the ARPES experiment, we observe the surface band structures and the temperature evolution of surface band splitting, which qualitatively agree with our first-principles calculations. The agreement between the first principles calculations and ARPES experimental observation is evidence for topological surface states in topological materials. We also note that our prediction that the non-magnetic phase is trivial, and thus no topological surface states. Consistently, no surface states are observed above T_N . However, the band folding due to magnetic order results in band inversion so that a topological gap opens in parts of the Brillouin zone. The agreement between theory and experimental findings both above and below T_N is evidence that the surface states observed in ARPES are topological.

3. The author should also include the units of k_x and k_y in Fig. 4b?

Reply: We have added the units (\AA^{-1}) and the scale to Fig. 4b in the revised manuscript.

Reviewer #3 (Remarks to the Author):

Huang et al. have conducted interesting experiments on a rare-earth mononictide NdSb which is believed to exhibit complex magnetism interplaying with a topological band structure. The authors observed non-collinear antiferromagnetic order with multiple modulations using SPSTM. They also discovered a hidden spin-rotation transition which seems to be related to the splitting of surface states measured in ARPES. Using DFT calculations, the authors showed the magnetic orderings induce band folding in Brillouin zone, leading to the band gap opening in the inverted bulk band which provides room for the existence of the topological surface states. This manuscript establishes an interesting example of magnetism interplaying with topology in real materials, so I believe the manuscript has the potential for publication in Nature Communications. However, I have several questions that are not entirely clear to me within the present manuscript.

Reply: We thank the reviewer for a concise summary and positive evaluation of our work. Below we provide point-to-point replies to his/her comments/questions.

(1) How can the 1q-spin structure (Fig. 1e) result in the strip pattern in the SPSTM measurement? If I consider the spins of the top Nd atoms, I am not sure why q_b -AFM peaks

should be absent. Should I take into account the underlying Nd atoms or the top Sb atoms to understand the missing q_b -AFM peaks? If so, I don't agree with the authors regarding STM's ability to detect the spins of buried Nd atoms.

Reply: We thank the reviewer for the questions on SP-STM contrast. The reviewer is correct that q^B should also be present if one only considers spins of surface Nd atoms. Because of tunneling nature, SP-STM contrast comes from spin polarized LDOS of itinerant electrons on the surface. Thus, one should not expect SP-STM image would directly reflect the local moments of surface magnetic atoms. In NdSb, the $1q$ spin structure due to both surface and buried Nd atoms results in a stripe pattern of spin density modulation of bulk states, which is revealed by our SP-STM.

The reviewer is correct that SP-STM cannot directly detect the spins of buried Nd atoms. However, SP-STM can detect spin polarized contrast of electronic states on surface if they are polarized by the buried magnetic atoms. For example, in Ref [29, 30], the magnetic contrast in SP-STM is due to the magnetic order formed by the magnetic atoms (Fe or Ir) that are buried underneath the topmost non-magnetic atomic layer.

(2) What causes the anti-crossing in the folded bulk bands? Is it spin-orbit coupling? How the spin-rotation increases the anti-crossing gap? How does the increased anti-crossing gap split the surface band further?

Reply: Spin-orbital coupling results in an indirect band inversion, i.e., the Nd-5d band minimum at the X point is lower than a Sb-3p band maximum at the Γ point as shown in Fig. R1(a). The anti-crossing in the folded bulk bands is allowed by the reduced symmetry of the magnetic order which folds the Brillouin zone, as shown in Fig. R1.

The spin-rotation does slightly enhance the anti-crossing gap, which is proportional to the size of order moment. However, the main enhancement comes from the sharply increased of order moment below the spin-rotation transition as shown in Fig. 3f and Supplementary Information Fig. 7, assuming that the order moment is proportional to the observed spin modulation intensity. The scale of the surface bands splitting is set by the size of anti-crossing gap because of topological nature of the surface bands. Thus, the enhanced anti-crossing gap split the surface bands further.

(3) What drives the spin-rotation transition around 13 K in terms of energy?

Reply: DFT calculations show that all rotated structures are very close in energy, with an energy difference of less than $1\text{ meV}/\text{atom}$, for all angles. This suggests that fluctuations, coupled with the structural symmetry breaking could spontaneously drive spin rotation. The lattice instability, in the presence of SOC could be responsible for spin rotation. Interestingly, spin-rotation transition has also been observed in rare-earth orthoferrite (REFeO_3 , where RE is a rare-earth element) due to the competition between different types of magnetic interactions. (e.g. *Nat. Mat.* 11, 694–699

(2012) and *J. Appl. Phys.* 101, 123919 (2007)) Future studies such as high-resolution x-ray scattering might reveal the driving force of the spin-rotation transition.

(4) In Fig. 4a, it's unclear why the ARPES intensities are not repeated within the AFM Brillouin zone (indicated by blue dashed lines) if the Brillouin zone is folded in the AFM phase of NdSb. Can the authors provide a demonstration of how the bulk bands in the paramagnetic phase fold in the AFM phase? This clarification could be beneficial for understanding the mechanisms of band inversions as a result of Brillouin zone folding.

Reply: We thank reviewer for the constructive question and suggestion. The ARPES intensities are not equivalent in different AFM Brillouin zones because of the matrix element effect of the ARPES experiment. For example, in Fig. 4a the bulk bands intensity at Γ_1 is stronger than that at X. The same inequivalent intensities around different zone centers have also been observed in the previous ARPES experiments. [Sakhya et. al, *Phys. Rev. B* 106, 235119 (2022), Kushnirenko et. al, *Phys. Rev. B* 106, 115112 (2022), Li et. al, *npj Quantum Mater.* 8, 22 (2023), Honma et. al., *Phys. Rev. B* 108, 115118 (2023)]. Honma et. al., *Phys. Rev. B* 108, 115118 (2023) also point out this observation is due to the matrix element effect.

Following reviewer's suggestion, we added a figure in supplementary information (Fig. 14) for the demonstration of band folding and topological gap opening which is also shown in Fig. R1.

(5) In Fig. 4, NbSb should be NdSb in several places.

Reply: We thank the reviewer for pointing out the typos. They are corrected in the revised version.

List of changes in the revised manuscript

1. Cited and discussed the latest ARPES works in the discussion of the ARPES results in Page 6.
2. Added the scale and unit to Fig. 4b and corrected the typos “NbSb” to “NdSb” in the caption.
3. Added Fig. S14 in the Supplementary Information to illustrate the process of band folding and the hybridization gap opening.
4. Added explanation of band folding and hybridization gap opening in the second paragraph in Page 7.
5. Added discussion in the first paragraph in Page 8 to emphasize that spin-rotation transition enhances order moments and therefore increases the hybridization gap and the surface band splitting.
6. Move all the Extended figures to Supplementary Information to adhere to the format instruction of Nature Communication.

[The revised text is colored in red in the highlighted pdf.]

Reviewers' Comments:

Reviewer #1:

Remarks to the Author:

The authors have addressed the points I raised before, so now I can recommend the publication of the work.

Reviewer #2:

Remarks to the Author:

I read the authors response with great interest. I found that the authors responded the technical question satisfactorily. However, I still feel that this manuscript lacks the novelty required for Nature Communication publication. I believe that the manuscript fits better for a more specialized journal.

Reviewer #3:

Remarks to the Author:

The authors have addressed my concerns, clarifying that the topological phase transition in NdSb could be induced by the band folding associated with the AFM phase transition. Consequently, I am now convinced. Therefore, I recommend this manuscript for publication in NC.